# Adiponectin is essential for lipid homeostasis and survival under insulin deficiency and promotes β-cell regeneration

Risheng Ye[1], William L Holland[1], Ruth Gordillo[1], Miao Wang[2], Qiong A Wang[1], Mengle Shao[1], Thomas S Morley[1], Rana K Gupta[1], Andreas Stahl[3], Philipp E Scherer[1,2,4]*

[1]Touchstone Diabetes Center, Department of Internal Medicine, University of Texas Southwestern Medical Center, Dallas, United States; [2]Hamon Center for Therapeutic Oncology Research, University of Texas Southwestern Medical Center, Dallas, United States; [3]Department of Nutritional Sciences and Toxicology, University of California Berkeley, Berkeley, United States; [4]Department of Cell Biology, University of Texas Southwestern Medical Center, Dallas, United States

*For correspondence: philipp. scherer@utsouthwestern.edu

Competing interests: The authors declare that no competing interests exist.

**Abstract** As an adipokine in circulation, adiponectin has been extensively studied for its beneficial metabolic effects. While many important functions have been attributed to adiponectin under high-fat diet conditions, little is known about its essential role under regular chow. Employing a mouse model with inducible, acute β-cell ablation, we uncovered an essential role of adiponectin under insulinopenic conditions to maintain minimal lipid homeostasis. When insulin levels are marginal, adiponectin is critical for insulin signaling, endocytosis, and lipid uptake in subcutaneous white adipose tissue. In the absence of both insulin and adiponectin, severe lipoatrophy and hyperlipidemia lead to lethality. In contrast, elevated adiponectin levels improve systemic lipid metabolism in the near absence of insulin. Moreover, adiponectin is sufficient to mitigate local lipotoxicity in pancreatic islets, and it promotes reconstitution of β-cell mass, eventually reinstating glycemic control. We uncovered an essential new role for adiponectin, with major implications for type 1 diabetes.

## Introduction

Adiponectin is an adipocyte-derived hormone exerting pleiotropic beneficial effects on metabolism (*Ye and Scherer, 2013*). Increased circulating adiponectin improves the metabolic flexibility of adipose tissue and confers systemic tolerance to obesity (*Asterholm and Scherer, 2010*). Under normal physiological conditions, adiponectin promotes plasma lipid clearance (*Combs et al., 2004*; *Qiao et al., 2008*).

Paradoxically however, even though many important functions have been attributed to this circulating factor in mice and humans (*Shetty et al., 2009*; *Turer and Scherer, 2012*), adiponectin is not essential for life under normal physiological conditions. Genetic deletion of adiponectin in rodents leads to mild or moderate insulin resistance, which is exacerbated upon high-fat diet challenge (*Kubota et al., 2002*; *Maeda et al., 2002*). Adiponectin is required for PPARγ agonist-mediated improvements in insulin sensitivity (*Nawrocki et al., 2006*). Based on many published studies, adiponectin function under normal physiological function is dispensable, and it starts to play a more prominent role under hyperglycemic and, most importantly, dyslipidemic conditions.

**eLife digest** Fat tissue is essential for health. Fat cells store energy and release it when it is needed; they also release hormones that are important for the health of our heart and for regulating our metabolism. One of these hormones, adiponectin, helps cells to remove fat molecules from the bloodstream. This allows the body to maintain appropriate cholesterol levels and prevents fatty build-ups from blocking blood vessels, which is associated with heart disease. Adiponectin also helps cells respond to insulin, a hormone that regulates blood sugar levels, and thus helps to prevent diabetes.

Despite this hormone's important roles in health, mice that lack adiponectin can thrive under normal conditions. Adiponectin becomes essential, however, when blood sugar or fat levels are considerably altered. For example, when mice without adiponectin are fed a high fat-content diet, they become insulin-resistant. Moreover, certain diabetes drugs that boost insulin sensitivity only work if adiponectin is present in the body.

Adiponectin helps to keep the β-cells that produce insulin alive. In patients with diabetes, β-cells slowly die, and this leads to a poor insulin response and an imbalance in the amount of fats and sugars in the cells. This is toxic to the β-cells; and as more β-cells die, less insulin is produced to control sugar levels, and the condition worsens. Adiponectin appears to protect the β-cells against this vicious cycle, but the details of how it does so are unclear.

Ye et al. used a mouse model in which β-cells were destroyed to see what adiponectin does when insulin is in short supply. When insulin levels were extremely low, adiponectin was found to help one type of fat tissue absorb fat molecules from the bloodstream, which reduced blood cholesterol levels. It also protects fat cells from being destroyed when insulin levels are low. Ye et al. also found that mice that lack both insulin and adiponectin lose excessive amounts of fat tissue and develop high blood cholesterol levels, which lead to death.

Increasing adiponectin levels in insulin-deficient mice, however, improves their blood cholesterol levels and protects β-cells from being destroyed. This allows the β-cells to begin regenerating. As the β-cells regenerate, the insulin-deficient mice developed better control over their blood sugar.

Many people with type-1 diabetes (which is caused by their own immune system destroying their β-cells) currently rely on insulin injections and restricted diets to manage their condition. Ye et al.'s findings might lead to new strategies to restore β-cell production and blood sugar control; as such these findings will have important implications for the management of type-1 diabetes.

The failure of insulin-producing β-cells is a hallmark of the pathophysiology of both type 1 and type 2 diabetes. The ongoing loss of β-cells under these conditions is associated with a failure to effectively regenerate β-cell mass. This can be attributed not only to the low capacity for replication and differentiation (*Bouwens and Rooman, 2005*) but also the detrimental cytotoxic environment that the cells are exposed to due to the dysregulation of the balance between insulin and glucagon (*Robertson, 2009*; *Unger and Cherrington, 2012*). Insulin deficiency results in hyperglycemia and hyperlipidemia, both of which trigger β-cell glucotoxicity and oxidative stress (*Poitout and Robertson, 2008*), endoplasmic reticulum (ER) stress (*Fonseca et al., 2009*), and lipotoxicity (*Kusminski et al., 2009*). Tissue culture data suggested that adiponectin regulates β-cell viability (*Brown et al., 2010*; *Holland et al., 2011*; *Rao et al., 2012*). Recently, we demonstrated that adiponectin protects β-cells against lipotoxicity and apoptosis, both in cultured cells and in vivo (*Holland et al., 2011*).

In this study, we took advantage of the PANIC-ATTAC transgenic mouse model (*Wang et al., 2008*). After extensive β-cell ablation, adiponectin becomes essential for survival. In the context of insulin deficiency, the lack of adiponetin aggravates the lipoatrophy and hyperlipidemia to lethal levels. The critical role of adiponectin in maintaining minimal lipid homeostasis is recapitulated in the streptozotocin-treated (STZ) mouse model. Specifically, adiponectin is required for lipid uptake into subcutaneous white adipose tissue. Under normal conditions, the action of adiponectin can be mediated through enhanced lipoprotein lipase activity and intracellular fatty acid translocation. However, under insulinopenic conditions, the primary adiponectin-mediated effect relies on enhanced insulin sensitivity and endocytic activity. While insulin deficiency and widespread loss of β-cells lead to augmented intracellular lipotoxicity, adiponectin overexpressing mice effectively overcome the resulting

intracellular lipotoxicity in β-cells by ameliorating lipid metabolism and thereby paving the way for β-cell mass recovery.

Our findings reveal a novel role of adiponectin as a housekeeping protein under insulinopenic conditions, and augmentation of adiponectin is sufficient to promote β-cell regeneration.

## Results

### Adiponectin is required for minimal lipid homeostasis and survival in PANIC-ATTAC mice

To investigate the physiological role of adiponectin under insulinopenic conditions, we crossed adiponectin null mice (*Nawrocki et al., 2006*) to the homozygous PANIC-ATTAC background (*Wang et al., 2008*). The PANIC-ATTAC transgene allows us to eliminate a defined number of β-cells. Starting with similar β-cell mass (*Figure 1A* and *Figure 1—figure supplement 1*), 8-week old male homozygous PANIC-ATTACs with adiponectin wild type (*P-Adn$^{+/+}$*) or knockout (*P-Adn$^{-/-}$*) were treated with the same high-dose of dimerizer AP20187 to induce caspase-8-mediated apoptosis. 2 weeks after the initial dimerizer administration, the insulin positive areas of *P-Adn$^{+/+}$* and *P-Adn$^{-/-}$* mice decreased down to <15% of their starting levels and further decreased down to <10% around 10 weeks post β-cell ablation (*Figure 1A* and *Figure 1—figure supplement 1*). Different from our previous studies (*Wang et al., 2008*; *Holland et al., 2011*), we have used a fairly stringent ablation protocol to obtain a very high level of β-cell loss in both genotypes. In this setting, both the *P-Adn$^{+/+}$* and *P-Adn$^{-/-}$* mice showed sustained glucose levels above ~500 mg/dl (*Figure 1B*). Their fasting insulin levels decreased to <8% of the euglycemic wild-type (*WT*) controls after parallel dimerizer treatment (*Figure 1C*). With such intensive β-cell ablation and insulin deficiency, both the *P-Adn$^{+/+}$* and *P-Adn$^{-/-}$* mice were severely glucose intolerant (*Figure 1D*), and their glucose-stimulated insulin secretion (GSIS) was abolished (*Figure 1E*).

To our surprise, under the insulinopenic conditions, *P-Adn$^{-/-}$* mice showed a dramatically lower survival rate than *P-Adn$^{+/+}$* mice (*Figure 1F*). Only 33% of the adiponectin null mice survived 9 weeks post dimerizer, with a median survival of 8.4 weeks, while 87% of the *P-Adn$^{+/+}$* mice remained alive. Adiponectin overexpressing mice (*Combs et al., 2004*) crossed into the homozygous PANIC-ATTAC background (*P-Adn$^{Tg/+}$*) show a similar survival rate as *P-Adn$^{+/+}$* mice. This fivefold increased mortality in *P-Adn$^{-/-}$* mice was associated with an extreme deterioration in lipid metabolism. Their triglyceride levels (6.8 ± 1.0 mM) were 60% higher than the levels in *P-Adn$^{+/+}$* mice (*Figure 1G*). More strikingly, the circulating ketone bodies in the adiponectin null mice reached an aberrantly high level (0.77 ± 0.15 mM), which was sixfold higher than the *WT* level and 3.5-fold higher than the *P-Adn$^{+/+}$* level (*Figure 1H*). Prior to the aggravated hyperlipidemia, *P-Adn$^{-/-}$* mice demonstrated a significant decrease in fat mass, to a critically low level of <5% body weight (*Figure 1I* and *Figure 1—figure supplement 2*). We observed no significant difference in food intake between *P-Adn$^{-/-}$* and *P-Adn$^{+/+}$* mice (*Figure 1—figure supplement 3*). These data indicate that adiponectin is essential for lipid homeostasis and survival in the absence of insulin.

### Adiponectin is critical for lipid metabolism in STZ-induced insulinopenic diabetes

To further elucidate the critical role of adiponectin in lipid metabolism under insulinopenia, we treated adiponectin knockout mice (*Adn$^{-/-}$*) and the *WT* controls (*Adn$^{+/+}$*) with a high dose of streptozotocin (STZ) as an alternate approach to destroy β-cells. Consistent with the observations in the PANIC-ATTAC model (*Figure 1I*), STZ-treated adiponectin null mice had significantly lower adipose tissue mass than *WT* mice (*Figure 2A* and *Figure 2—figure supplement 1*). The deterioration in lipid metabolism in the STZ-treated *Adn$^{-/-}$* mice was also apparent as judged by the increase in circulating triglyceride levels in fed mice (2.3-fold) (*Figure 2B*), overnight fasted mice (3.6-fold) (*Figure 2—figure supplement 2A*), and following an oral triglyceride load (*Figure 2C*). Non-esterified fatty acids (NEFAs) levels were also higher in *Adn$^{-/-}$* mice (1.4-fold increase) (*Figure 2—figure supplement 2B*). The plasma lipoprotein fractionation of STZ-treated *Adn$^{-/-}$* mice displayed elevated triglyceride and cholesterol content in VLDL (*Figure 2D,E*). In an attempt to rescue this phenotype not only genetically, but also using a recombinant protein approach, exogenous administration of adiponectin induced a lowering of circulating triglycerides in STZ-treated *Adn$^{-/-}$* mice (*Figure 2—figure supplement 3*).

We wanted to determine what the underlying mechanisms are for the exacerbated lipid metabolism in STZ-treated adiponectin knockout mice. We addressed whether changes in hepatic lipid

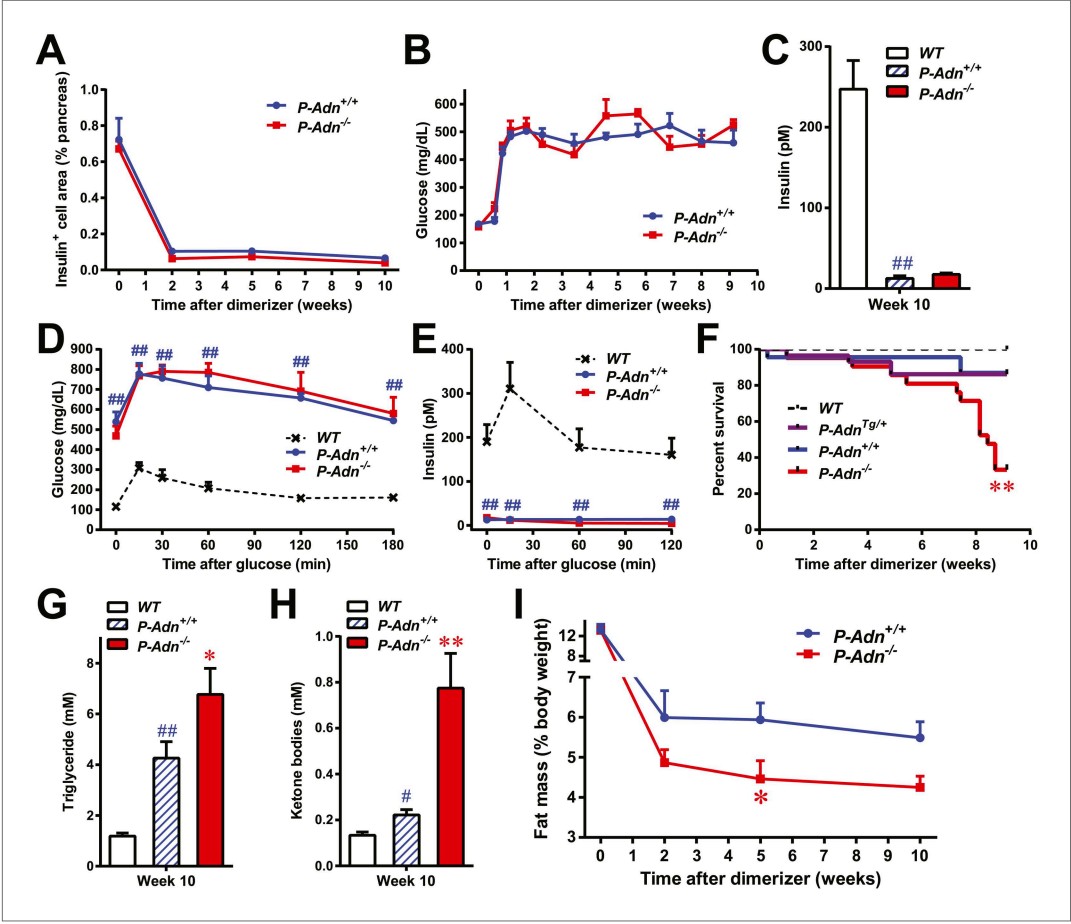

**Figure 1**. Adiponectin is required for minimal lipid homeostasis and survival in PANIC-ATTAC mice. Mice with four different genotypes (wildtype [*WT*]; homozygous PANIC-ATTAC with adiponectin wildtype [*P-Adn$^{+/+}$*], adiponectin knockout [*P-Adn$^{-/-}$*], or transgenic adiponectin overexpressing mice [*P-Adn$^{Tg/+}$*]) were exposed to dimerizer. (**A**) Quantitation of insulin-immunostained cell area, normalized to total pancreas area. n = 3–6 mice per condition. Source files are available in *Figure 1—source data 1*. (**B**) Fasting blood glucose. n ≥ 8 mice per condition. (**C–E**) At 10 weeks after dimerizer treatment: (**C**) fasting blood insulin. (**D**) Blood glucose and (**E**) plasma insulin during an oral glucose tolerance test. (**F**) Survival curve. n = 25 (*WT*), 29 (*P-Adn$^{Tg/+}$*), 23 (*P-Adn$^{+/+}$*) and 21 (*P-Adn$^{-/-}$*) mice. (**G** and **H**) Blood triglycerides (**G**) and total ketone bodies (**H**) in mice under fed status. (**I**) Fat mass presented as percentage of mouse body weight. n ≥ 6 mice per condition unless specified. Data are presented as the mean ± SEM. *p < 0.05, **p < 0.01 for *P-Adn$^{-/-}$* vs *P-Adn$^{+/+}$*. #p < 0.05, ##p < 0.01 for *P-Adn$^{+/+}$* vs *WT*.

The following source data and figure supplements are available for figure 1:

**Source data 1**. Source files for insulin-positive cell area quantitation.

**Figure supplement 1**. Sustained β-cell ablation in PANIC-ATTAC mice.

**Figure supplement 2**. Body composition of PANIC-ATTAC mice.

**Figure supplement 3**. Food intake of PANIC-ATTAC mice.

secretion may be the underlying reason. We treated the mice with the lipoprotein lipase inhibitor tyloxapol (WR-1339) and monitored serum lipid accumulation, a classical test to assess VLDL secretion. Compared to *WT* control mice, adiponectin null mice demonstrated only a minor trend towards an increase in serum triglyceride content (*Figure 2F*) and rate of accumulation (*Figure 2—figure supplement 4*). We also examined a panel of metabolic gene expression in the livers of STZ-treated *Adn$^{+/+}$* and *Adn$^{-/-}$* mice and observed no significant changes (*Figure 2—figure supplement 5*).

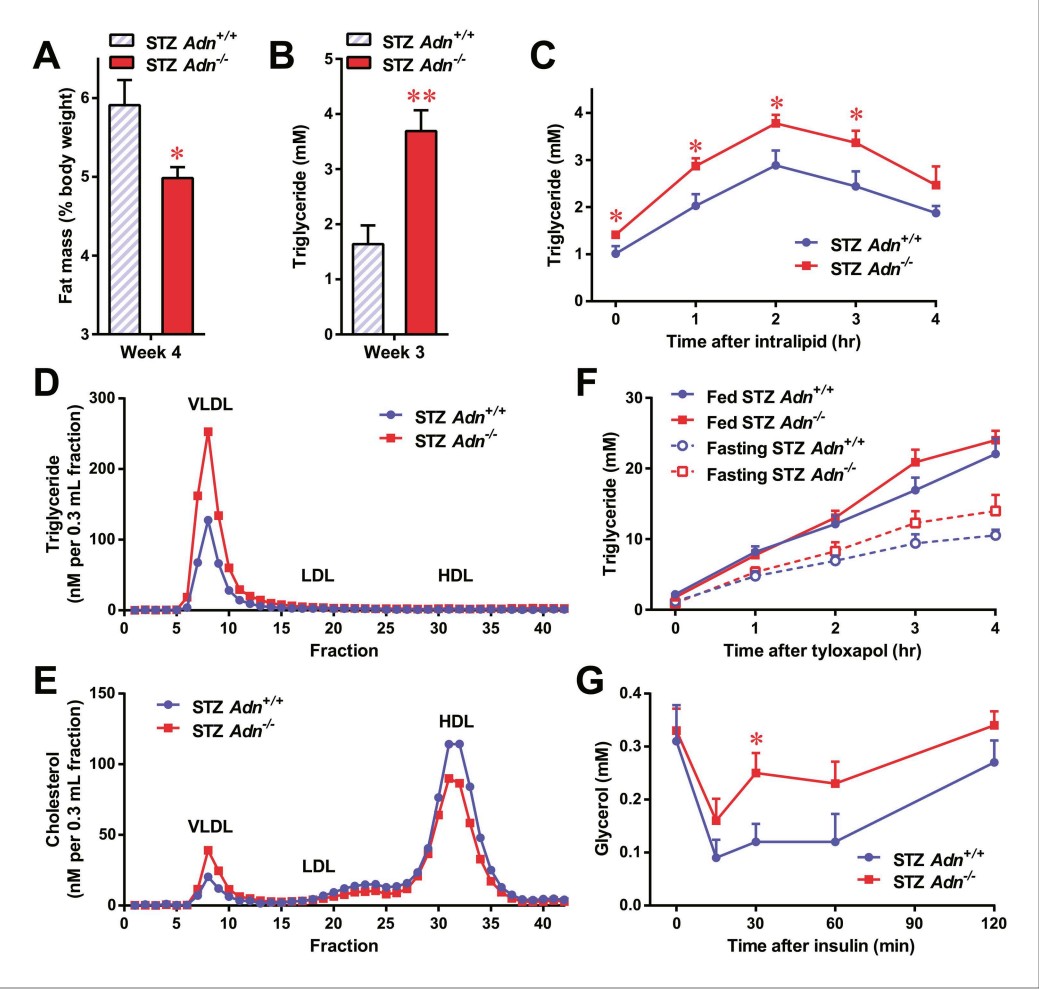

**Figure 2**. Adiponectin is essential for lipid metabolism in STZ-treated insulinopenic mice. Adiponectin wild-type (*Adn*[+/+]) and knockout (*Adn*[−/−]) mice were treated with streptozotocin (STZ) to eliminate pancreatic β-cells and became hyperglycemic within 2 days after administration. (**A**) Fat mass 4 weeks after STZ treatment. (**B**) Fed triglyceride levels 3 weeks after STZ treatment. (**C**) Blood triglycerides during oral triglyceride tolerance test in mice 2 weeks after STZ treatment. (**D** and **E**) Pooled plasma samples from *Adn*[+/+] and *Adn*[−/−] mice (n = 3) were subjected to FPLC lipoprotein fractionation. Triglyceride (**D**) and cholesterol (**E**) contents of the 0.3 ml fractions were assayed. (**F**) Blood triglycerides after tail vein injection of tyloxapol. 2 weeks after STZ treatment, mice were fed or fasted for 4 hr before tyloxapol administration. (**G**) Blood glycerol after intraperitoneal injection of insulin (0.1 mU/g BDW) in mice 3 days after STZ treatment. n ≥ 7 mice per condition unless specified. Data are presented as the mean ± SEM. *p < 0.05, **p < 0.01 for *Adn*[+/+] vs *Adn*[−/−].

The following figure supplements are available for figure 2:

**Figure supplement 1**. Body composition of STZ-treated mice.

**Figure supplement 2**. Serum lipids in overnight fasted STZ-treated mice.

**Figure supplement 3**. Recombinant adiponectin rescues triglyceride clearance in STZ-treated adiponectin null mice.

**Figure supplement 4**. Hepatic secretion rates of triglycerides.

**Figure supplement 5**. Hepatic expression of metabolic genes.

**Figure supplement 6**. Serum triglycerides and NEFAs after low-dose insulin administration.

Insulin regulates fat metabolism in adipocytes by both enhancing FFA uptake and inhibiting lipolysis (*Holm et al., 2000*). Does adiponectin play a role in suppressing lipolysis under conditions of low insulin? To address this question, we administered a low dose of insulin to STZ-treated mice (0.1 mU/g body weight), and measured circulating glycerol levels under these conditions. This small amount of insulin substantially reduced serum glycerol (*Figure 2G*), while it only minimally affected circulating triglycerides (*Figure 2—figure supplement 6A*) and NEFAs (*Figure 2—figure supplement 6B*). Intriguingly, compared to *WT* mice, adiponectin null mice were resistant to the action of insulin, with minimal impact on glycerol levels (*Figure 2G*). This suggests adiponectin is critical for insulin-mediated suppression of lipolysis under insulinopenic conditions. The enhanced lipolysis in STZ-treated adiponectin null mice might, at least in part, account for the reduced fat mass seen in the $Adn^{-/-}$ mice (*Figure 2A*).

## Adiponectin is required for lipid uptake specifically in subcutaneous white adipose tissues in STZ-treated mice

Yet another site of action could be at the level of plasma lipid clearance. We examined whole body and tissue-specific uptake of circulating triglycerides using $^3$H-triolein. No apparent differences between *WT* and adiponectin null mice were noted prior to STZ treatment. In contrast, post STZ treatment, adiponectin null mice showed a 39% lower whole body clearance rate of labeled triolein compared to *WT* mice (*Figure 3—figure supplement 1A*). Among the nine tissues examined, we observed significant differences of triolein uptake only in the subcutaneous white adipose tissue (WAT, 50% lower in STZ-treated null mice than in STZ-treated *WT* mice) (*Figure 3A,B*). Consistent with a primary site of action on subcutaneous fat pads, we have reported that after long-term high-fat diet exposure, adiponectin promotes preferentially subcutaneous WAT expansion (*Asterholm and Scherer, 2010*). We also partitioned the tissue-specific $^3$H-triolein uptake into incorporated (*Figure 3—figure supplement 1B*) vs oxidized lipids (*Figure 3—figure supplement 1C*). The difference in triolein uptake in subcutaneous WAT was primarily the result of a reduced level of incorporation. In agreement with the $^3$H-triolein uptake assay, histological analysis of adiponectin knockouts demonstrated a major reduction of adipocyte size in subcutaneous WAT and trends towards smaller cell size in brown adipose tissue (BAT). However, no such reduction was found in gonadal WAT (*Figure 3C*). We further confirmed the essential role of adiponectin in lipid uptake employing the PANIC-ATTAC model. 3 weeks post dimerizer, $P-Adn^{Tg/+}$, $P-Adn^{+/+}$ and $P-Adn^{-/-}$ mice were subjected to oral gavage of BODIPY-labeled fatty acids and examined for fluorescence signal in subcutaneous WAT by confocal microscopy. Adiponectin overexpression dramatically enhanced BODIPY signal in adipocytes, while adiponectin nulls showed a significant reduction (*Figure 3D* and *Figure 3—figure supplement 2*).

To determine whether the defective lipid uptake leads to lipoprotein accumulation in subcutaneous WAT, we examined apolipoproteins A1, B, and E (ApoA1, ApoB, and ApoE, major components of high-, low-, and intermediate-density lipoproteins, respectively) in situ by immunofluorescence (*Figure 3E–G*). Compared to the euglycemic wild-type controls, STZ-treated animals showed elevated signals for apolipoproteins. Interestingly, most of the ApoA1 and ApoE signals, as well as part of the ApoB signal, located adjacent to endomucin, a marker for venous and capillary endothelial cells, reflecting apolipoproteins in circulation. The endomucin signal itself demonstrated increased vascular density in subcutaneous WAT of STZ-treated lipoatrophic animals, which was also observed in gonadal WAT (*Figure 3—figure supplement 3–5*). Furthermore, adiponectin knockout-induced ApoA1 accumulation was significant in subcutaneous WAT and BAT, but not in gonadal WAT (*Figure 3E* and *Figure 3—figure supplement 3*). Compared to STZ-treated wild-type mice, adiponectin null mice showed trends towards an increase in ApoB signal in both subcutaneous and gonadal WATs (*Figure 3F* and *Figure 3—figure supplement 4*). As for the ApoE signal, there was a trend towards an increase in subcutaneous WAT, but not in gonadal WAT (*Figure 3G* and *Figure 3—figure supplement 5*). Collectively, these data suggest an overall accumulation of apolipoproteins in the local circulation of adipose tissues of STZ-treated mice, which is exacerbated by adiponectin depletion predominantly in subcutaneous WAT.

Insulin promotes lipid storage in adipose tissue via stimulating the intracellular insulin signaling cascades leading to enhanced extracellular lipoprotein lipase activity. A low dose of insulin treatment (0.2 mU/g body weight) markedly suppressed the hormone-sensitive lipase (HSL) serine-660 phosphorylation, a marker positively associated with lipolysis, by 57% and induced Akt serine-473 phosphorylation by 8.6-fold in subcutaneous WAT of STZ-treated wild-type mice. However, STZ-treated

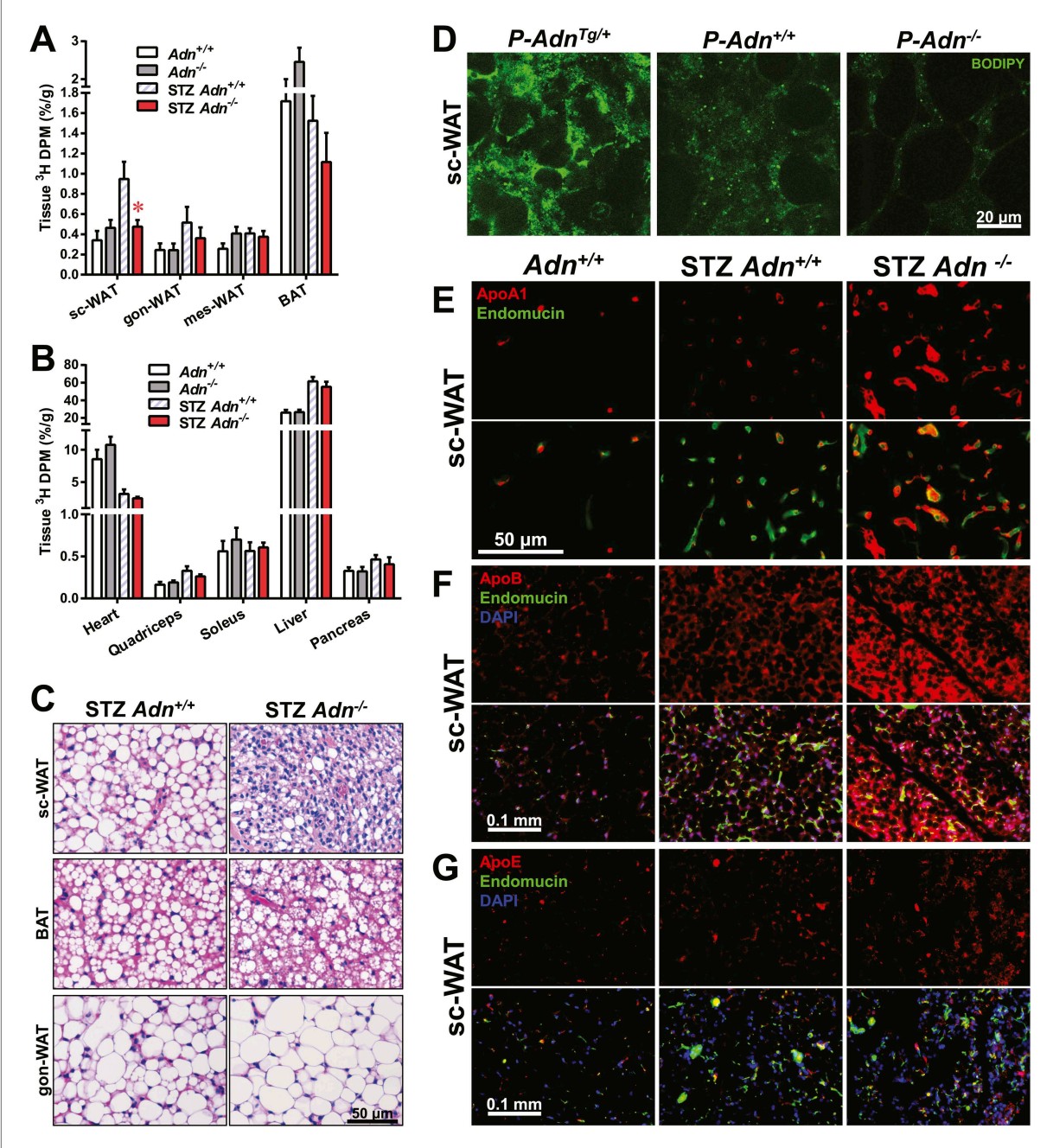

**Figure 3**. Adiponectin is critical for subcutaneous white adipose tissue lipid uptake in mice after β-cell ablation. (**A** and **B**) Total $^3$H radioactivity in adipose tissues at the end of the $^3$H-triolein injection experiment. Radioactivity is calculated as percentage of input, and normalized against tissue weight. Sc: subcutaneous. Gon: gonadal. Mes: mesenteric. WAT: white adipose tissue. BAT: brown adipose tissue. Mice were either controls or used 3 weeks after STZ treatment. n ≥ 6 mice per condition. Data are presented as the mean ± SEM. *p < 0.05 for $Adn^{+/+}$ vs $Adn^{-/-}$. (**C**) Representative H&E stains of sections from subcutaneous WAT, BAT, and gonadal WAT. (**D**) Representative confocal microscopy of BODIPY fluorescence signal in whole-mount subcutaneous WAT. PANIC-ATTAC mice were at 3 weeks post dimerizer and subjected to an oral gavage of BODIPY-labeled fatty acids (2 μg/g BDW) 3 hr before tissue collection. (**E**–**G**) Representative immunofluorescence co-stain of apolipoproteins A1 (**E**), B (**F**), or E (**G**) (red), and endomucin (green) in subcutaneous WAT.

The following figure supplements are available for figure 3:

**Figure supplement 1**. Tissue-specific catabolism of $^3$H-triolein.

*Figure 3. Continued on next page*

*Figure 3. Continued*

**Figure supplement 2**. Quantitation of BODIPY signal in subcutaneous WAT.

**Figure supplement 3**. ApoA1 immunofluorescence in gonadal WAT and BAT with quantitation.

**Figure supplement 4**. ApoB immunofluorescence in gonadal WAT with quantitation.

**Figure supplement 5**. ApoE immunofluorescence in gonadal WAT with quantitation.

adiponectin nulls showed a blunted response in HSL inhibition (41%), and an abolished Akt activation (13%), as compared to the same animal prior to insulin injection (*Figure 4A* and *Figure 4—figure supplement 1*). The unchanged post-heparin lipoprotein lipase activity (*Figure 4—figure supplement 2*) is unlikely to be a major contributing factor to the impaired lipid uptake in the STZ-treated adiponectin null animals. The fatty acid translocase CD36 (*Figure 4B*) and fatty acid transport protein 1 (FATP1) (*Figure 4C*) are also unlikely contributors, both of which show comparable distributions between genotypes. Expression of Scavenger Receptor Class B Member 1 (SR-B1), a high-density lipoprotein (HDL) receptor, was enhanced in STZ-treated adiponectin knockouts (*Figure 4D*), which could reflect a compensatory response for the HDL accumulation in circulation (*Figure 3E* and *Figure 3—figure supplement 3C*).

Employing transmission electron microscopy, we observed a twofold increase in the linear density of vesicular structures associated with the plasma membrane of the subcutaneous adipocytes in the wild-type mice after STZ treatment. In contrast, this upregulation of vesicle density was abolished in the adiponectin nulls (*Figure 4E,F*). These changes were consistent with the $^3$H-triolein uptake data, which measured the lipid uptake capacity per gram tissue (*Figure 3A*). Interestingly, STZ treatment enhanced mitochondrial density in both $Adn^{+/+}$ and $Adn^{-/-}$ mice (*Figure 4—figure supplement 3*), consistent with the $^3$H-triolein oxidation capacity (*Figure 3—figure supplement 1C*). Caveolin-1, one of the major components in the plasma membrane and trans-Golgi network, was detectable in the majority of subcutaneous adipocytes in wildtype mice after STZ treatment. In contrast, in STZ-treated adiponectin null mice, the caveolin-1 signal was almost completely depleted in subcutaneous adipocytes, and the majority of the signal was associated with the endomucin-positive endothelial cells (*Figure 4G*) under those conditions. These changes were not observed in either the euglycemic controls or the gonadal WAT (*Figure 4G,H*).

These findings suggest that under conditions of limited insulin availability, adiponectin plays an important role in potentiating insulin sensitivity, supporting endocytic activity for triglyceride, and promoting lipid storage specifically in subcutaneous WAT. The lack of adiponectin exacerbates lipoatrophy and hyperlipidemia, and this may be due–at least in part–to the diminished insulin signaling and the selective loss of the caveolin-1 complex.

## Adiponectin overexpression improves systemic lipid metabolism in PANIC-ATTAC mice

We subsequently wanted to investigate whether adiponectin can rescue the dyslipidemia brought about by insulin deficiency in the form of a genetic gain-of-function mutant that overexpresses adiponectin. Adiponectin overexpressing mice (*Combs et al., 2004*) were crossed into the homozygous PANIC-ATTAC background ($P$-$Adn^{Tg/+}$) and treated with the same high-dose of dimerizer as the $P$-$Adn^{+/+}$ and $P$-$Adn^{-/-}$ mice to induce β-cell apoptosis. The adiponectin transgenic mice sustained higher levels of plasma adiponectin than the wild-type mice, also in the absence of insulin (*Figure 5A*). Dimerizer treatment led to a moderate decrease in whole-body fat mass by week 2. From that point onward, the adiponectin transgenic mice showed a complete recovery in fat mass by week 5 and sustained normal fat mass through week 10, at which point the experiment was stopped (*Figure 5B* and *Figure 5—figure supplement 1*). Early post β-cell ablation, $P$-$Adn^{Tg/+}$ mice already showed significant improvements during a triglyceride tolerance test compared to $P$-$Adn^{+/+}$ mice (*Figure 5C*). Furthermore, the baseline serum triglyceride levels were significantly lower in $P$-$Adn^{Tg/+}$ mice compared to $P$-$Adn^{+/+}$ mice at all stages (*Figure 5D*), indicative of the powerful lipid-lowering effects of adiponectin on the clearance of circulating triglycerides. Subsequently, $P$-$Adn^{Tg/+}$ mice restored their serum ketone bodies to a

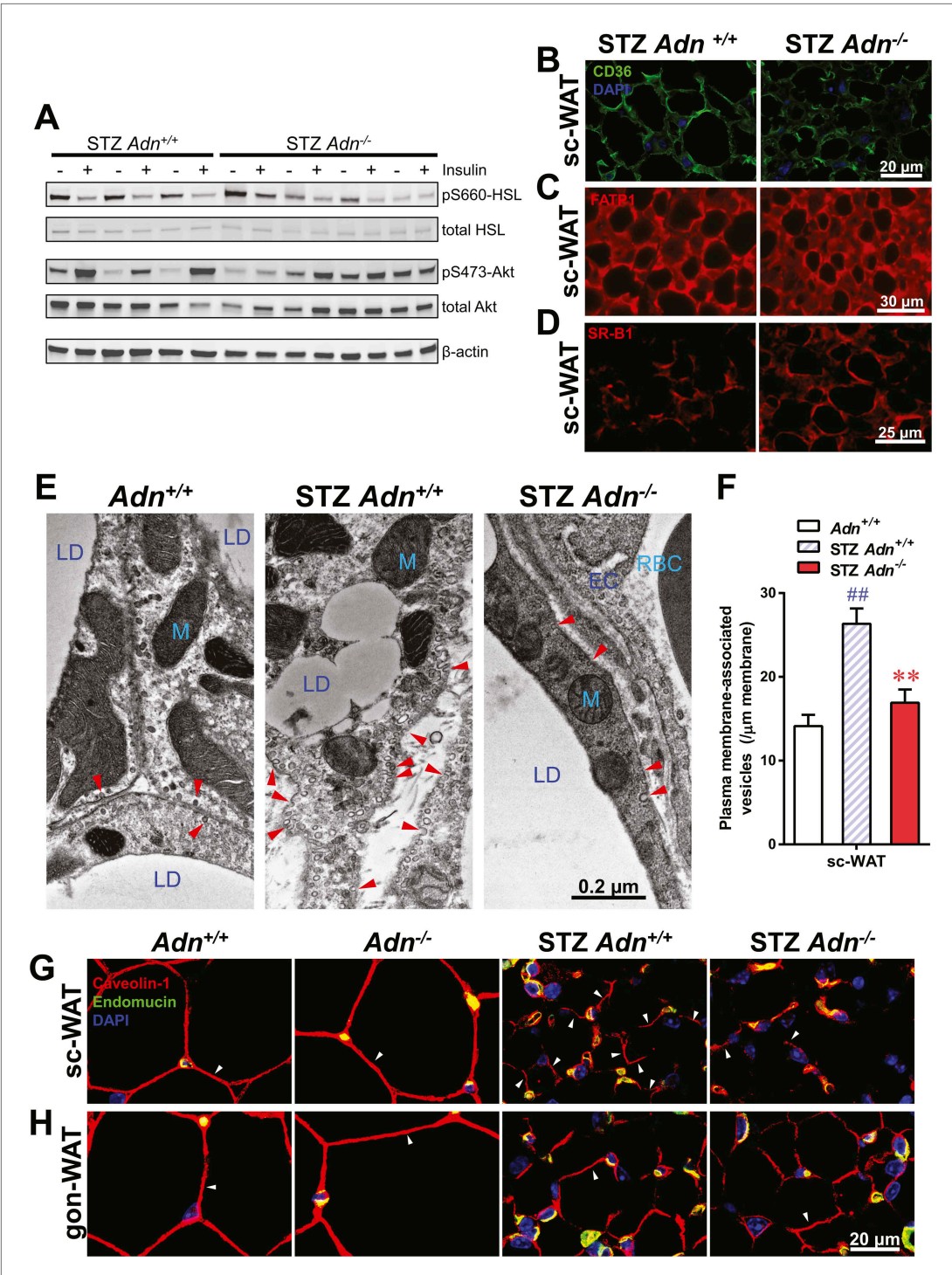

**Figure 4**. Adiponectin is important for caveolar structures and Caveolin-1 expression in subcutaneous white adipose tissue of STZ-treated mice. (**A**) Western blots of insulin signaling molecules, hormone-sensitive lipase (HSL, serine 660 phosphorylated and total) and Akt (serine 473 phosphorylated and total) in inguinal subcutaneous WAT. 1 week after STZ treatment, overnight fasted mice were subjected to a tail vein injection of insulin (0.2 mU/g BDW). Every two adjacent lanes represent the paired fat pads from an individual mouse, before (−) or 5 min after (+) insulin injection. (**B–D**) Representative immunofluorescence of CD36 (**B**), fatty acid transport protein 1 (FATP1) (**C**), and HDL receptor SR-B1 (**D**) on subcutaneous WAT of STZ-treated mice. (**E** and **F**) Transmission electron microscopy of subcutaneous white adipocytes. Vesicles <10 nm from the plasma membrane were defined as 'PM associated'. (**E**) Representative fields of adipocyte plasma membranes. LD: lipid droplet. M: mitochondrion. EC: endothelial cell. RBC: red blood cell. Arrowheads: examples of vesicles associating with plasma membrane. (**F**) Quantitation of plasma membrane-associated vesicles normalized against
*Figure 4. Continued on next page*

*Figure 4. Continued*

membrane length. n ≥ 17 fields per condition. (**G** and **H**) Representative confocal co-immunofluorescence of caveolin-1 (red) and endomucin (green) on subcutaneous (**G**) and gonadal (**H**) WAT. Arrowheads: examples of caveolin-1 signal on adipocytes. Data are presented as the mean ± SEM. **p < 0.01 for *Adn⁻/⁻* vs *Adn⁺/⁺* after STZ treatment. ##p < 0.01 for STZ-treated vs untreated *Adn⁺/⁺*.

The following figure supplements are available for figure 4:

**Figure supplement 1**. Quantitation of insulin effect on phosphorylation of HSL and Akt.

**Figure supplement 2**. Lipoprotein lipase activity.

**Figure supplement 3**. Quantitation of mitochondrial density on electron microscopic images.

**Figure supplement 4**. Quantitation of caveolin-1 immunofluorescence.

level (0.12 ± 0.02 mM) comparable to unchallenged *WT* controls (0.13 ± 0.01 mM) and significantly lower than both the *P-Adn⁺/⁺* and *P-Adn⁻/⁻* mice (***Figure 5E***). We also observed trends towards a decrease in NEFAs in adiponectin transgenic mice (***Figure 5—figure supplement 2***).

Our findings suggest that increasing circulating adiponectin is sufficient to improve systemic lipid metabolism and promote fat mass recovery in the context of insulinopenic diabetes. The lack of available insulin action uncovers adiponectin's importance for lipid homeostasis in the absence of insulin.

## Adiponectin mitigates islet lipotoxicity in PANIC-ATTAC mice

We have previously shown that adiponectin enhances INS-1 β-cell survival by reducing accumulation of ceramides, a family of lipotoxic sphingolipids (***Holland et al., 2011***). To investigate the in vivo roles of adiponectin on lipotoxicity in β-cells, we measured different sub-species of sphingolipids in pancreatic islets isolated from *WT*, *P-Adn⁺/⁺*, and *P-Adn^{Tg/+}* mice 5 weeks post β-cell ablation. For most individual ceramide species as well as the total ceramide content, we observed a trend in diabetic *P-Adn⁺/⁺* islets towards increased ceramides as compared to islets from euglycemic *WT* mice and a concomitant drop of ceramides in the *P-Adn^{Tg/+}* islets (***Figure 6A***). The same trends applied to dihydro-ceramides, which are ceramide biosynthetic precursors (***Figure 6B***). The pro-survival lipids, sphingosine-1-phosphate and sphinganine-1-phosphate were below the detection limit in all islet preparations, and sphingoid bases were not significantly altered by adiponectin (data not shown). Among the hexosyl-ceramides, lactosyl-ceramides displayed the same pattern of change as ceramides overall (***Figure 6C***), while no significant changes were observed in glucosyl-ceramides (***Figure 6D***). As the major storage pool of sphingolipids, most of the sphingomyelins also reproduced the trends seen in ceramides among the three genotypes (***Figure 6E***). Taken together, our data suggest increased levels of lipotoxic species in *P-Adn⁺/⁺* islets, which may account, at least in part, for the further loss of β-cells from week 5 to week 10 post the initial insult (***Figure 1A***). In contrast, adiponectin transgenic mice reduced the lipotoxic sphingolipid content to levels seen in the non-diabetic *WT* mice. It is likely that this phenomenon accounts for the enhanced pro-survival effects on the β-cells.

In addition to lipotoxic lipids, hyperglycemia could impose potent negative effects via glucotoxicity and subsequent pro-apoptotic oxidative and ER stress in β-cells. We wondered whether adiponectin counterbalances these negative cellular events as well. However, as judged by immunohistochemistry on pancreatic islets, no apparent difference was observed in 8-hydroxyguanosine (an oxidative stress marker), or CHOP (a marker for pro-apoptotic ER stress) (data not shown).

## Adiponectin promotes β-cell regeneration in PANIC-ATTAC mice

As in the *P-Adn⁺/⁺* and *P-Adn⁻/⁻* mice, the high-dose dimerizer treatment resulted in >85% ablation of β-cell mass in *P-Adn^{Tg/+}* mice at week 2 (***Figure 7A*** and ***Figure 7—figure supplement 1***). In contrast to the prolonged β-cell loss in the *P-Adn⁺/⁺* and *P-Adn⁻/⁻* mice from week 5 to week 10 (***Figure 1A***), the *P-Adn^{Tg/+}* β-cell area showed a significant recovery in β-cell mass during this period. The transgenic mice restored their islet mass to 29% of the *WT* controls that did not suffer β-cell ablation (***Figure 7A*** and ***Figure 7—figure supplement 1***). Importantly, the recovery of β-cell mass in *P-Adn^{Tg/+}* mice was preceded by improvements in both the systemic lipid metabolism (***Figure 5B–D***) and the local islet

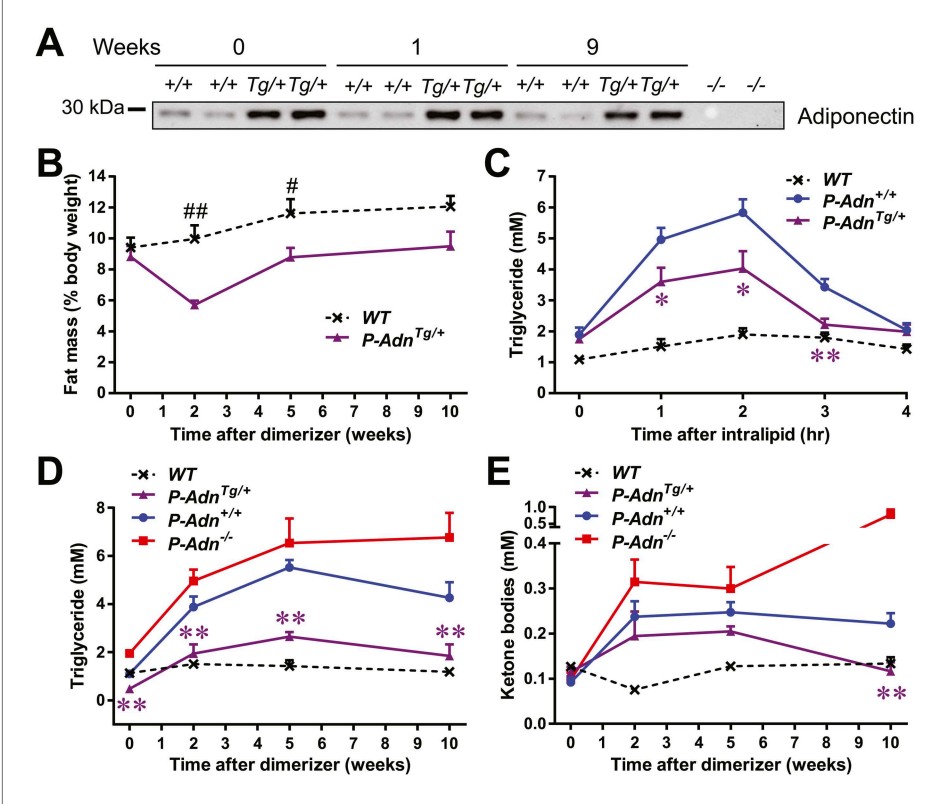

**Figure 5**. Adiponectin improves systemic lipid metabolism in PANIC-ATTAC mice. (**A**) Serum samples were collected from *P-Adn^{+/+}* (+/+) and *P-Adn^{Tg/+}* (*Tg/+*) mice, 0, 1, and 9 weeks after initial dimerizer treatment and subjected to Western blotting for adiponectin. Equal volume of serum was loaded for each lane. Serum samples from adiponectin knockout mice (−/−) were included as negative controls. (**B**) Fat mass presented as percentage of mouse body weight (BDW). (**C**) Blood triglyceride during oral triglyceride tolerance test in mice 2 weeks after initial dimerizer treatment. (**D** and **E**) Plasma triglyceride (**D**) and total ketone bodies (**E**) in mice under fed status. Data of *WT*, *P-Adn^{+/+}*, and *P-Adn^{−/−}* at week 10 were presented in *Figure 1G,H*. For B to E, n ≥ 5 mice per condition. Data are presented as the mean ± SEM. *p < 0.05, **p < 0.01 for *P-Adn^{Tg/+}* vs *P-Adn^{+/+}*. #p < 0.05, ##p < 0.01 for *WT* vs *P-Adn^{Tg/+}*.

The following figure supplements are available for figure 5:

**Figure supplement 1**. Body composition of PANIC-ATTAC mice.

**Figure supplement 2**. Fed NEFA levels in PANIC-ATTAC Mice.

lipotoxicity (*Figure 6*), potentially supporting a causal relationship of adiponectin-mediated lipid improvements leading to islet mass recovery.

With the partially restored β-cell mass, *P-Adn^{Tg/+}* mice significantly down-regulated their fasting blood glucose level to 199 ± 21 mg/dl (*Figure 7B*). The restoration to euglycemia was observed in both female (*Figure 7—figure supplement 2A*) and aged male *P-Adn^{Tg/+}* mice (*Figure 7—figure supplement 2B*), supporting a sex- and age-independent effect of adiponectin in β-cell regeneration. We further examined the adiponectin-driven improvements in glucose metabolism and β-cell function in the *P-Adn^{Tg/+}* mice. The mice displayed significant improvements in glucose tolerance (*Figure 7C*), fasting insulin, and in vivo GSIS (*Figure 7D* and *Figure 7—figure supplement 3*). To directly test the β-cell function during the recovery stage, we isolated pancreatic islets from the *P-Adn^{+/+}* and *P-Adn^{Tg/+}* mice and subjected them to in vitro GSIS assays, with non-treated, euglycemic mice as controls (*Figure 7—figure supplement 4A*). The *P-Adn^{Tg/+}* islets showed an ~fivefold increase in insulin secretion under both basal conditions and upon exposure to elevated glucose levels (*Figure 7E*). This improvement could be attributed to increases in both overall islet insulin content

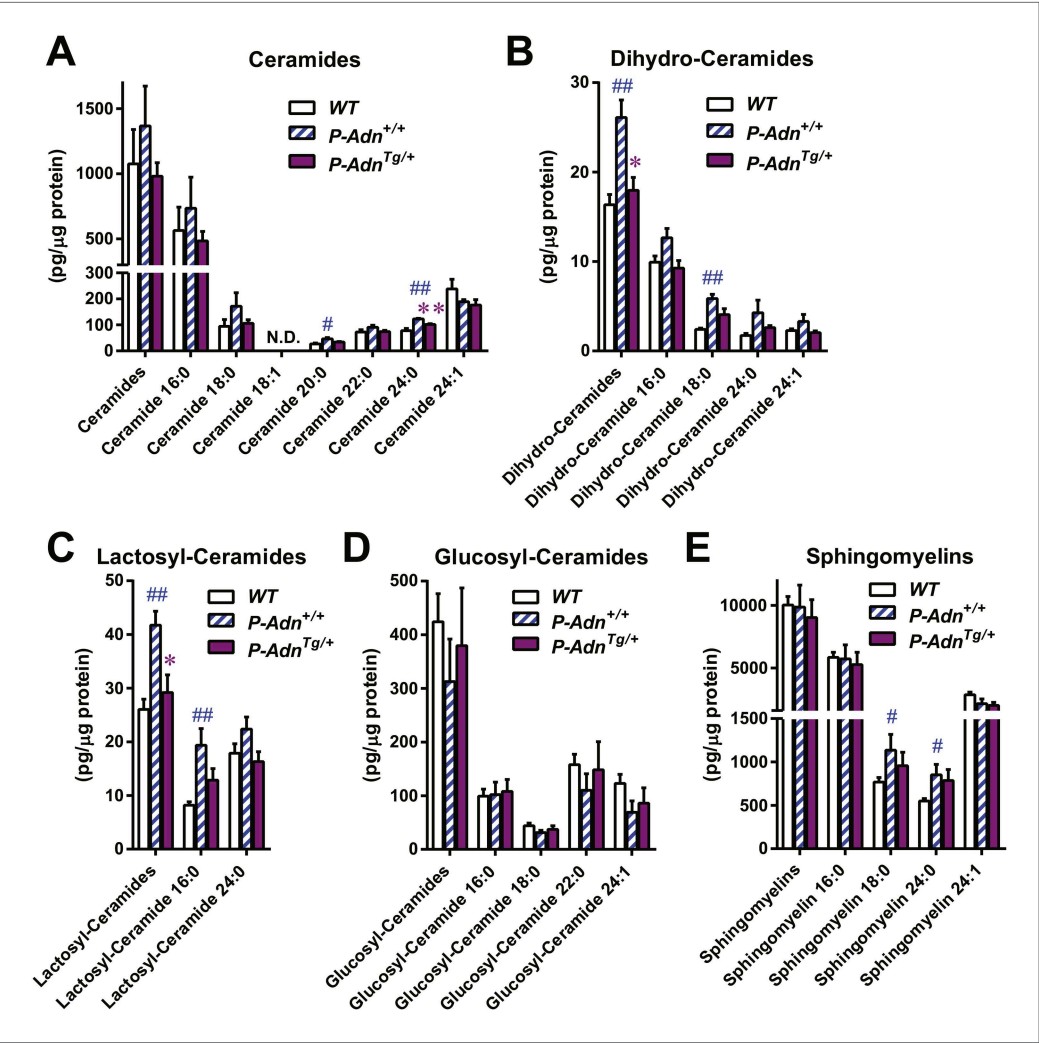

**Figure 6**. Adiponectin reduces lipotoxic sphingolipids in regenerating PANIC-ATTAC islets. Sphingolipids were assayed by mass spectrometry in pancreatic islets isolated from mice 5 weeks after initial dimerizer treatment and normalized against protein content of islet samples. n = 3–5 samples per genotype. Sphingolipid species are categorized as (**A**) ceramides, (**B**) dihydro-ceramides, (**C**) lactosyl-ceramides, (**D**) glucosyl-ceramides, and (**E**) sphingo-myelins. In every panel, the first group of columns from the left represents the sums of all species combined. Data are presented as the mean ± SEM. *p < 0.05, **p < 0.01 for *P-Adn*$^{Tg/+}$ vs *P-Adn*$^{+/+}$. ##p < 0.01 for *P-Adn*$^{+/+}$ vs *WT*. N.D.: not detected.

(1.7-fold, *Figure 7—figure supplement 4B*) and exocytic activity (2.2-fold, *Figure 7—figure supplement 4C*).

Enhanced proliferative activity was observed prior to the major regeneration of β-cells in the *P-Adn*$^{Tg/+}$ mice. Five weeks post ablation, *P-Adn*$^{Tg/+}$ mice showed higher ratios of Ki-67-positive (*Figure 7F* and *Figure 7—figure supplement 5A*) and BrdU-positive (*Figure 7G* and *Figure 7—figure supplement 5B*) cells in insulin-positive cells than seen in the islets of *P-Adn*$^{+/+}$ or *WT* mice. The potent adiponectin-driven improvements in systemic lipid metabolism and amelioration of local lipotoxicity may be an important facilitative component towards β-cell proliferation and may contribute to the eventual recovery of islet mass.

## Discussion

In this study, we uncover for the first time a pathological condition under which adiponectin is indispensable for survival. When insulin levels decrease by >90% after intensive β-cell loss, adiponectin

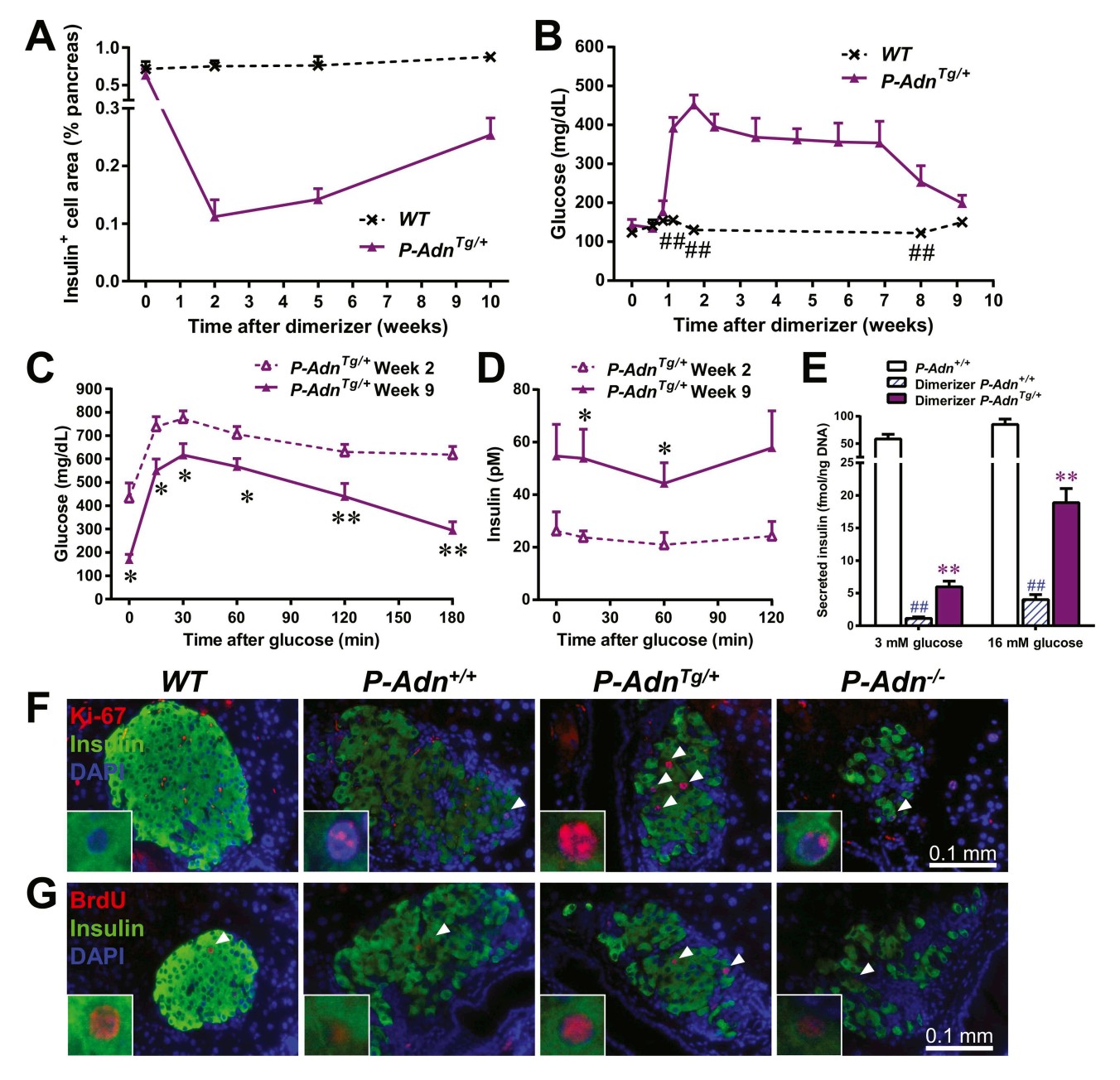

**Figure 7**. Adiponectin promotes β-cell recovery in PANIC-ATTAC mice. (**A**) Quantitation of insulin-immunostained cell area normalized to total pancreas area. n = 3–7 mice per condition. Source files are available in **Figure 7—source data 1**. (**B**) Fasting blood glucose. n ≥ 5 mice per condition. ##p < 0.01. (**C** and **D**) At 2 and 9 weeks after initial dimerizer treatment, *P-Adn*$^{Tg/+}$ were subjected to an oral glucose tolerance test. Plasma glucose (**C**) and insulin (**D**) were determined. n ≥ 5 mice per condition. *p < 0.05, **p < 0.01. (**E**) Pancreatic islets were isolated from dimerizer-treated *P-Adn*$^{+/+}$ and *P-Adn*$^{Tg/+}$ mice at the recovery stage, with untreated *P-Adn*$^{+/+}$ mice as controls. Insulin secretion from islets under basal (3 mM) or stimulating (16 mM) glucose concentrations was measured and normalized against the DNA content of islets. n = 3–8 samples per condition. **p < 0.01 for dimerizer-treated *P-Adn*$^{+/+}$ vs *P-Adn*$^{Tg/+}$. ##p < 0.01 for dimerizer-treated vs untreated *P-Adn*$^{+/+}$. (**F** and **G**) Representative immunofluorescence (red) of Ki-67 (**F**) and BrdU (**G**) stains in pancreatic islets of mice 5 weeks after initial dimerizer treatment, merged with insulin (green) and DAPI (blue). Arrowheads: Ki-67$^+$ Insulin$^+$ (**F**) or BrdU$^+$ Insulin$^+$ (**G**) cells. Insets: representative nuclear signal (purple). For BrdU incorporation, mice were subjected to an i.p. injection of BrdU at the dose of 100 μg/g BDW 6 hr before sacrifice and tissue processing. Data are presented as the mean ± SEM.

The following source data and figure supplements are available for figure 7:

**Source data 1**. Source files for insulin-positive cell area quantitation.

*Figure 7. Continued on next page*

*Figure 7. Continued*

**Figure supplement 1**. Adiponectin promotes β-cell recovery in PANIC-ATTAC mice.

**Figure supplement 2**. Adiponectin restores glycemic control in female and aged PANIC-ATTAC mice.

**Figure supplement 3**. Adiponectin enhances in vivo GSIS in PANIC-ATTAC mice.

**Figure supplement 4**. Adiponectin enhances insulin content and in vitro GSIS in islets isolated from PANIC-ATTAC mice.

**Figure supplement 5**. Quantitation of proliferating β-cells.

ensures the minimal homeostasis for lipid metabolism. Consistent with insulin action promoting lipid storage in adipose tissue, adiponectin exerts a similar effect on lipid uptake, but does so by distinct mechanisms. First, adiponectin mediates lipid uptake specifically in subcutaneous WAT, but not in epididymal WAT. Hepatic VLDL secretion and fatty acid metabolism are unchanged. Second, adiponectin regulates neither lipoprotein lipase activity under insulinopenic conditions nor the intracellular translocation of fatty acids. Rather, caveolin-1 is reduced, thereby reducing endocytosis of lipids. This is in agreement with our previous reports on caveolin-1 knockout mice, which show significantly lower body weight and fat mass than wild-type controls under both regular chow and high-fat diet regimen (*Razani et al., 2002*; *Wernstedt Asterholm et al., 2012*). Moreover, our adiponectin knockout mice with insulin deficiency recapitulate the lipodystrophic phenotypes reported in caveolin-1 null mice, including impaired triglyceride clearance, hyper-triglyceridemia, adipocyte hypotrophy, without a change in lipoprotein lipase activity (*Razani et al., 2002*). Third, adiponectin potentiates insulin signaling and the suppression on lipolysis mediated by the trace levels of insulin present. It is tempting to speculate that adiponectin exerts autocrine/paracrine actions on subcutaneous adipose as a hormone, motivating future studies on signaling pathways and transcriptome/proteome analysis under insulin-openic conditions. The generation of mouse models allowing for conditional elimination of both adiponectin receptors adipoR1 and adipoR2 should be insightful. In cases where both insulin and adiponectin are depleted, fat mass becomes critically low, accompanied by exceedingly high triglycerides and ketone bodies in circulation, resulting in major mortality.

Our findings also underscore adiponectin as a messenger for the crosstalk between adipose tissue and pancreatic β-cells, especially under insulinopenic conditions. Massive pancreatic β-cell failure leads to hypoinsulinemia and dyslipidemia (*Dunn, 1992*). The ensuing aggravated lipotoxicity further impairs β-cell function and survival (*Lupi et al., 2002*; *Kusminski et al., 2009*). Our findings demonstrate that adiponectin can disrupt this vicious cycle at multiple levels and, in so doing, promote potent regenerative effects on functional β-cell mass. Adiponectin improves lipid storage in adipose tissue and improves systemic lipid metabolism. These general improvements in dyslipidemia may contribute, at least in part, to the reduction of β-cell lipotoxicity in the adiponectin overexpressing mouse, as reflected by ceramide measurements. The reduced level of local lipotoxicity may mediate β-cell survival and proliferation. Overexpression of adiponectin is sufficient to regenerate β-cells and regain glycemic control. Taken together, our data highlight the link between lipid metabolism and β-cell maintenance and identify adiponectin as a key mediator of this process. This does not exclude a direct effect of adiponectin on β-cells, since adiponectin is avidly binding to β-cells in vivo and exerts potent cytoprotective effects on β-cells under these conditions (*Holland et al., 2011*).

The major impediments for effective β-cell regeneration may not only relate to the inherently low proliferation rate of β-cells, but also due to the systemic highly lipotoxic environment due to severe hyperlipidemia, both in type 1 diabetes as well as late stage type 2 diabetes. In vitro experiments have provided strong evidence that lipotoxicity can impair β-cell function and survival (*Maedler et al., 2001*; *Rakatzi et al., 2004*; *Hoppa et al., 2009*; *Holland et al., 2011*). Unger and colleagues demonstrated in rats that pancreatic islets transplanted into the hepatic portal area were destroyed by the local hyperlipidemic environment (*Lee et al., 2007*) and islets from Zucker diabetic fatty (ZDF) rats are subject to lipotoxic destruction via ceramide (*Shimabukuro et al., 1998*). Poitout and colleagues have established that wild-type islets similarly develop ceramide-induced impairments in islet function under hyperlipidemic/hyperglycemic conditions (*Kelpe et al., 2003*). In the PANIC-ATTAC model, the

prolonged β-cell loss in adiponectin wild-type and adiponectin null mice is unlikely due to prolonged exposure to dimerizer, which has a half-life in mice of ~5 hr. Rather, this is the result of glucotoxicity, lipotoxicity, oxidative stress, and/or ER stress. In the PANIC-ATTAC islets, we did not detect any differences in oxidative stress and ER stress markers between adiponectin wild-type and transgenic adiponectin overexpressing mice (data not shown). However, we did see differences at the level of lipotoxic intermediates as judged by the results from the islet ceramide assays that revealed a reduction of lipotoxic ceramides as well as their precursors and derivatives. Ultimately, this improved microenvironment allows for increased recovery of islet mass.

## Materials and methods

### Mice

Mice were bred and maintained on a 12-hr dark/light cycle, with *ad libitum* access to water and regular chow diet (#5058; LabDiet, St. Louis, MO). The strains were generated and previously described by our laboratory: PANIC-ATTAC (*Wang et al., 2008*), adiponectin overexpressing mice (*Combs et al., 2004*), and adiponectin null mice (*Nawrocki et al., 2006*). All mice were maintained on a FVB background. Body composition including fat mass and lean mass were measured with a Bruker Minispec mq10. Food intake was recorded on individually housed mice for 5 consecutive days. All protocols for mouse use and euthanasia were reviewed and approved by the Institutional Animal Care and Use Committee of the University of Texas Southwestern Medical Center (UTSW).

### Genotyping PCR

Primer pairs for genotyping PCR were: 5′-GAAAGTGCCCAAACTTCAGAGCATTAGG-3′ and 5′-AACTGAGATGTCAGCTCATAGATGGGGG-3′ for PANIC-ATTAC; 5′-GTTCCTCTTAATCCTGCCCAGTC-3′ and 5′-CCCGGAATGTTGCAGTAGAACTTG-3′ for adiponectin transgenic; 5′-TTGGACCCCTGAACTTGCTTCACACC-3′ and 5′-GGATGCGGTGGGCTCTATGGCTTC-3′ for adiponectin knockout allele; 5′-TTGGACCCCTGAACTTGCTTCACACC-3′ and 5′-TCCTGAGTTCAATTCCCAGCACCCAC-3′ for adiponectin wildtype allele. The PCR program was: 95°C for 5 min, followed by 35 cycles of 95°C for 15 s, 62°C for 30 s, and 72°C for 30 s, ended with 72°C for 3 min.

### Dimerizer administration

Mice were subjected to six daily intraperitoneal (i.p.) injections of the dimerizer AP20187 (Clontech, Mountain View, CA) at the dose of 0.5 µg/g body weight (BDW)/day. The dimerizer was stored at −20°C as 12.5 g/l solution in 100% ethanol, and freshly diluted in 2% Tween 20 with 10% polyethylene glycol 400 before injection.

### Streptozotocin (STZ) administration

Mice were fasted for 6 hr and subjected to a single i.p. injection of streptozotocin (#S1030; STZ, Sigma, St. Louis, MO) at the dose of 135 µg/g BDW. STZ was stored at −20°C as powder and freshly diluted in ice-cold sodium citrate buffer (0.1 M, pH 4.5) before injection.

### Immunohistochemistry

Mice were euthanized by cervical dislocation following isoflurane anesthesia. Tissues were immediately collected and fixed overnight in 10% buffered formalin. Afterward, tissues were rinsed with 50% ethanol for three times, embedded in paraffin blocks by the University of Texas Southwestern Medical Center Molecular Pathology Core, and sliced for 5-µm sections. For BrdU incorporation, mice were subjected to an i.p. injection of BrdU (100 µg/g BDW) 6 hr before sacrifice.

Primary antibodies and dilution for immunostaining or immunofluorescence were: insulin (#A0564; 1:500; Dako, Carpinteria, CA), glucagon (#18-0064; 1:500; Invitrogen, Grand Island, NY), apolipoprotein A1 (#ab20453; 1:100; Abcam, Cambridge, MA), endomucin (#sc-65495; Santa Cruz Biotechnology, Santa Cruz, CA), apolipoprotein B (#ab20737; 1:50, Abcam), apolipoprotein E (#sc-6384; 1:100, Santa Cruz Biotechnology), CD36 (#NB400-144; 1:100; Novus Biologicals, Littleton, CO), FATP1 (1:25, Dr Andrea Stahl), SR-B1 (#NB400-104; 1:50; Novus), Caveolin-1 (#610493; 1:100; BD Biosciences, San Jose, CA), BrdU (#MCA2060; 1:50; AbD Serotec, Raleigh, NC), and Ki-67 (#M7249; 1:50; Dako).

For β-cell area quantitation, at least four sections per mouse pancreas, >50 µm apart from each other, were immunostained for insulin. Whole slides were scanned as color images with an Epson

Expression 10,000 XL photo scanner at a resolution of 2400 dpi. The insulin-positive brown area and the total pancreas area were quantitated with Adobe Photoshop, with a tolerance of 64 and 24, respectively. Immunofluorescence sections were examined with a Zeiss Axio Observer Z1 inverted microscope or a Leica TCS SP5 confocal microscope. Electronic/optical settings for image acquisition and parameters for linear digital processing were consistent among samples within the same experiment. Fluorescence intensity and area were quantitated with ImageJ.

### Assay of metabolites

Mice were fed *ad libitum* or fasted for 6 hr unless indicated otherwise. Blood was collected from a tail nick with a heparinized Microhematocrit capillary tube (#22-362-566; Fisher Scientific, Pittsburgh, PA) for plasma or a plain capillary (# 22-362-574; Fisher Scientific) for serum. Glucose was assayed with PGO enzymes (#P7119; Sigma) plus o-dianisidine (#F5803; Sigma). In serum samples, triglyceride was assayed with Infinity Triglycerides Liquid Stable Reagent (#TR22421; Thermo Scientific, Waltham, MA), NEFAs with HR Series NEFA-HR (2) (#999-34691; #995-34791; #991-34891; #993-35191; Wako, Richmond, VA), glycerol with Free Glycerol Reagent (#F6428; Sigma), and total ketone bodies with Autokit Total Ketone Bodies (#415-73301; #413-73601; Wako). ELISA kits were used to determine insulin (#EZRMI-13K; Millipore, Billerica, MA), and C-peptide2 (Millipore EZRMCP2-21K). For lipoprotein fractionation, fresh plasma was pooled and subjected to FPLC and assays for triglycerides and cholesterol at the UTSW Mouse Metabolic Phenotyping Core.

### In vivo tolerance tests

Mice were fasted for 4–6 hr before the tests unless indicated. Glucose tolerance test was initiated by oral gavage of dextrose (2 mg/g BDW), and plasma was collected at 0, 15, 30, 60, 120, and 180 min for glucose assay or at 0, 15, 60, 120 min for insulin and C-peptide2 assays. Triglyceride tolerance test was initiated by oral gavage of 20% Intralipid (10 µl/g BDW, #2B6022; Baxter, Deerfield, IL), and serum was collected at 0, 1, 2, 3, and 4 hr for triglyceride assay. Hepatic triglyceride secretion assay was initiated by tail vein injection of tyloxapol (0.5 mg/g BDW, #T0307; Sigma), and serum was collected at 0, 1, 2, 3, and 4 hr for triglyceride assays. The lipolysis suppression assay was initiated by i.p. injection of bovine insulin (0.1 mU/g BDW), and serum was collected at 0, 15, 30, 60, and 120 min for glycerol, triglyceride, and NEFA assays.

### Recombinant adiponectin administration

Mice were fasted for 4–6 hr and subjected to tail vein injection of recombinant adiponectin (2 µg/g BDW). Serum was collected at 0, 1, 2, 3, and 4 hr for triglyceride assays.

### RT-qPCR

cDNA was synthesized from the total RNA extract with the SuperScript II Reverse Transcriptase (#18064-014; Invitrogen) plus the RNaseOUT Recombinant Ribonuclease Inhibitor (#10777-019; Invitrogen). Quantitative real-time PCR (qPCR) was performed with Power SYBR Green PCR Master Mix (#4368708; Applied Biosystems, Carlsbad, CA) on a 7900HT Fast Real-Time PCR System (#4329001; Applied Biosystems), at least triplicate. See *supplementary file 1* for primer sequences.

### $^3$H-triolein chase

The $^3$H-triolein chase experiment was performed as previously described (*Kusminski et al., 2012*). The $^3$H-triolein (#NET431001MC; PerkinElmer, Waltham, MA) was dried under nitrogen flow, emulsified with 5 vol of 5% intralipid in PBS by 40 s sonication, and diluted 1:10 with PBS for tail vein injection. 10 µl of the injection solution were preserved for $^3$H scintillation counting to calculate input, and 200 µl per mouse were injected following 3–6 hr of fasting. At 1, 2, 5, 10, and 15 min after injection, 10 µl of tail blood was collected, immediately added into a scintillation vial containing 5 ml 3a70B complete counting cocktail (#111154; RPI Corp, Mount Prospect, IL), and shaken vigorously to disperse. Tissues were then immediately collected, weighed, immersed in 0.75 ml chloroform–methanol 2:1 mixture at 4°C overnight, and disrupted by bead vortex in a MagNA Lyser (#03358968001; Roche), at 5000 rpm, 30 s for two times. The content was then mixed with 0.5 ml 1 M CaCl$_2$ and centrifuge at 3000 rpm, 4°C for 30 min. The chloroform phase at the bottom, which contained the hydrophobic incorporated $^3$H-triolein, was transferred to a scintillation vial, air-dried completely in a fume hood, and mixed with 5 ml counting cocktail. The water–methanol phase supernatant, which

contained the hydrophilic oxidized $^3$H-triolein, was then transferred to a scintillation vial containing 5 ml counting cocktail, and shaken vigorously to mix. All the vials were counted 5 min for $^3$H scintillation in a Beckman Coulter LS6500 multi-purpose scintillation counter. Four-parameter double exponential decay regression was applied to calculate the whole body clearance rate based on the $^3$H activity in blood.

## In vivo uptake of BODIPY-labeled fatty acids

Mice were fasted for 4 hr and subjected to an oral gavage of BODIPY 500/510 $C_1$, $C_{12}$ fatty acids (2 µg/g BDW, #D3823; Molecular Probes). 3 hrs later, mice were euthanized and subcutaneous WAT was dissected. Tissues were immediately frozen in liquid nitrogen and stored at −80°C. Tissue pieces <2 mm were excised, mounted with fluorescence mounting medium (#S302380; Dako) on microscope slides with cover glasses, and examined for fluorescence with argon-ion laser excitation at 488 nm on a Leica TCS SP5 confocal microscope.

## Insulin signaling analysis

After overnight fasting, mice were anesthetized with isoflurane. The left inguinal subcutaneous WAT was excised. 5 mins after a tail vein injection of insulin (0.2 mU/g BDW), the right inguinal subcutaneous WAT was excised. Tissues were immediately frozen in liquid nitrogen and subsequently processed for protein lysate as previously described (*Ye et al., 2010a*). Protein separation and transfer were performed with 4–15% Mini-PROTEAN TGX Gels (#456-1086; Bio-Rad, Hercules, CA), Trans-Blot Turbo Mini Nitrocellulose Transfer Packs (#170-4158; Bio-Rad), and Trans-Blot Turbo Transfer Starter System (#170-4155; Bio-Rad). Immunoblots were imaged and quantitated with an IRDye 800CW IgG second antibody (#926-32211; LI-COR) and an Odyssey CLx infrared imaging system (LI-COR, Lincoln, NE). Primary antibodies included pSer660-HSL (#4126; 1:1000; Cell Signaling, Beverly, MA), HSL (#4107; 1:1000; Cell Signaling), pSer473-Akt (#9271; 1:1000; Cell Signaling), and Akt (#9272; 1:1000; Cell Signaling).

## Hepatic and lipoprotein lipase activity assay

Lipase activity was measured as previously described (*Razani et al., 2002*), with minor modifications. Briefly, pre- and post-heparin plasma was collected from mice before and after 15 min after tail vein injection of heparin (1.5 U/g BDW, #H3393; Sigma), respectively. Per 0.2 ml reaction, 10 µl of plasma was incubated with a triglyceride emulsion containing ~$10^7$ cpm/ml $^3$H-triolein (#NET431001MC; PerkinElmer) at 37°C water bath for 90 min for total lipase activity, or with 1.5 M NaCl for hepatic lipase activity. Reaction was terminated by addition of 3.25 ml methanol:chloroform:heptane (1.41:1.25:1) and 1.05 ml $K_2CO_3$ (pH 10.5). After vigorous agitation and centrifugation, 1 ml of the NEFA-containing aqueous phase was transferred and assayed for $^3$H radioactivity on a scintillation counter. Serial dilutions of a lipoprotein lipase (0, 10, 20, 50, and 100 ng, #L9656; Sigma) were parallel assayed as standard controls, and the lipase activities of plasma samples were calculated as equivalents of the standard lipase. The plasma lipoprotein lipase activity was calculated by subtracting the hepatic lipase activity from the total.

## Transmission electron microscopy

Tissues were processed at the UTSW Electron Microscopy Core Facility. Sections were examined with a JEOL 1200 EX electron microscope and photographed with a Sis Morada 11 MegaPixel side-mounted CCD camera.

## Serum adiponectin immunoblotting

Proteins in the same volume of serum were separated by 8% SDS-PAGE, transferred to nitrocellulose membranes (#162-0112; Bio-Rad), and blotted with a mouse adiponectin antibody (*Schraw et al., 2008*). The membrane was subsequently processed with the Odyssey imaging system (LI-COR).

## Isolation of pancreatic islets

Mice were euthanized by cervical dislocation following isoflurane anesthesia, and pancreatic islets were isolated as previously described (*Ye et al., 2010b*). Immediately after euthanasia, the major duodenal papilla of the mouse was blocked with a micro bulldog clamp. Ice-cold digestion solution, that is, Hank's Balanced Salt Solution (HBSS) with 0.1 g/l Liberase TL (#05401020001; Roche, Indianapolis, IN),

0.1 g/l DNase I (#10104159001; Roche), 2.5 mM HEPES, 8 mM glucose, 0.2% BSA, pH 7.2–7.4, was inject via the common bile duct into the pancreatic duct. The inflated pancreas was transferred into a scintillation vial with digestion solution on ice. Subsequently, the vial was incubated at 37°C water bath for 25 min and shaken vigorously to disperse the pancreas. The digested content was resuspended with ice-cold HBSS and settled on ice for 2–5 min, and supernatant was removed without disturbing the precipitate. The wash was repeated until the supernatant became clear, and the content was transferred into a 6-cm Petri dish with ice-cold HBSS plus 2.5 mM HEPES, 8 mM glucose, 0.2% BSA, pH 7.2–7.4. Under a dissection microscope, islets of Langerhans were picked with a 200-µl pipette to new dishes until free of exocrine pancreas content. The freshly isolated islets were either frozen immediately in liquid nitrogen followed by −80°C storage for sphingolipid assays, or transferred to culture medium for in vitro insulin secretion assays.

## Sphingolipid quantitation in pancreatic islets

Frozen mouse pancreatic islet samples (70–130 islets per sample) were homogenized in 0.5 ml aqueous buffer (25 mM HEPES, pH 6.8) using a sonic dismembrator system equipped with a 1/8-inch probe. The samples were kept on ice during the homogenization process. 50 µl of the homogenate was taken for protein determination by BCA assay, and the remaining sample was added to 2 ml of organic extraction mixture (isopropanol/ethyl acetate 15:85; vol:vol). Immediately afterward, 20 µl of internal standard solution (diluted 1:4 in ethanol, #LM6005; Avanti Polar Lipids, Alabaster, AL) was added. The mixture was vortexed and sonicated in ultrasonic bath for 10 min at 40°C. The samples were then allowed to reach room temperature and centrifuged at 3500 rpm in Sorvall Legend XTR (#75004521; Thermo Scientific). The supernatant was transferred to a new tube and the aqueous phase was re-extracted. Supernatants were combined and evaporated under nitrogen. The dried residue was reconstituted in 200 µl of HPLC solvent B (methanol/formic acid 99:1; vol:vol containing 5 mM ammonium formate) for LC-MS/MS analysis.

## In vitro glucose-stimulated insulin secretion

The freshly isolated pancreatic islets were cultured overnight in RPMI 1640 medium with 10% fetal bovine serum, 1% antibiotics, 8 mM glucose, and 0.2% BSA. 10–15 islets per sample were equilibrated in 1 ml secretion assay buffer (SAB) with 3 mM glucose for 1 hr and then transferred to 1 ml SAB with 3 mM glucose. After 1 hr incubation, 0.15 ml of SAB was frozen immediately in liquid nitrogen and stored at −80°C for insulin assay, and islets were transferred to 1 ml SAB with 16 mM glucose. After 1 hr incubation, 0.15 ml of SAB was sampled for insulin assay as previously, and islets were picked into 200 µl of 1 M acetic acid with protease inhibitors (#11836170001; Roche). Islets were subjected to 30 s sonication on ice, and 5 µl of the islet lysate were diluted 1:500 in 1 M acetic acid with protease inhibitors for insulin assay. 20 µl of 10 N NaOH was added to neutralize the remained islet lysate and then mixed with 300 µl of 100% isopropanol. DNA was precipitated after centrifugation at 13,000 rpm, 4°C for 15 min, rinsed briefly with 70% ethanol, and re-suspended in 100 µl 10 mM Tris-Cl (pH 8.5). DNA concentration was determined by SYBR green incorporation.

## Statistical analysis

Two-tailed student's $t$ test was applied for all pairwise comparisons. Kaplan–Meier survival curves were compared by the log-rank test. Statistical significance was accepted at $p < 0.05$.

## Acknowledgements

We thank the UTSW Molecular Pathology Core for tissue embedding and processing, especially John Shelton for advice on BODIPY detection in whole mount adipose tissue. We thank the UTSW Mouse Metabolic Phenotyping Core for metabolic assays. We thank Dr Roger H Unger for helpful discussions. We thank Steven Connell for administrative and technical support. This study was supported by the National Institutes of Health (grants R01-DK55758, R01-DK099110, and P01-DK088761-01 to PES) and the Juvenile Diabetes Research Foundation (JDRF 17-2012-36 to PES). RY was supported with a postdoctoral fellowship from the American Heart Association (11POST7240021) and a research fellowship from the Naomi Berrie Diabetes Center, Columbia University Medical Center. WLH was supported by the National Institutes of Health (grant K99-DK094973) and an American Heart Association Beginning Grant-in-Aid (12BGI-A8910006). The funding sources were not involved in study design, data collection and interpretation, or the decision to submit the work for publication.

# Additional information

## Funding

| Funder | Grant reference number | Author |
|---|---|---|
| National Institutes of Health | R01-DK55758 | Philipp E Scherer |
| Juvenile Diabetes Research Foundation International | 17-2012-36 | Philipp E Scherer |
| American Heart Association | 11POST7240021 | Risheng Ye |
| Columbia University | Naomi Berrie Diabetes Center, Research fellowship | Risheng Ye, Philipp E Scherer |
| National Institutes of Health | K99-DK094973 | William L Holland |
| American Heart Association | 12BGI-A8910006 | William L Holland |
| National Institutes of Health | R01-DK099110 | Philipp E Scherer |
| National Institutes of Health | P01-DK088761-01 | Philipp E Scherer |

The funders had no role in study design, data collection and interpretation, or the decision to submit the work for publication.

## Author contributions

RY, WLH, RG, MW, QAW, MS, Conception and design, Acquisition of data, Analysis and interpretation of data, Drafting or revising the article; TSM, Analysis and interpretation of data, Drafting or revising the article, Contributed unpublished essential data or reagents; RKG, PES, Conception and design, Analysis and interpretation of data, Drafting or revising the article; AS, Conception and design, Analysis and interpretation of data, Drafting or revising the article, Contributed unpublished essential data or reagents

## Ethics

Animal experimentation: All protocols for mouse use and euthanasia were reviewed and approved by the Institutional Animal Care and Use Committee of the University of Texas Southwestern Medical Center (#2010-0006).

# Additional files

## Supplementary file
• Supplementary file 1. Primers for RT-qPCR.

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
