## [Decision Letter]

Thank you for sending your work entitled “Adiponectin is essential for lipid homeostasis and survival under insulin deficiency and promotes β-cell regeneration” for consideration at *eLife*. Your article has been favorably evaluated by Fiona Watt (Senior editor) and 3 reviewers, two of whom are members of our Board of Reviewing Editors.

The Reviewing editor and the other reviewers discussed their comments before we reached this decision, and the Reviewing editor has assembled the following comments to help you prepare a revised submission.

The reviewers were in agreement that the work was exciting and potentially of interest to the readers of *eLife*. All of the reviewers felt that the discovery of new role for adiponectin is likely to have major impact. At the same time, there was also consensus that the mechanistic aspects of the work could be further developed. Given that the mechanism linking ceramide levels to beta-cell toxicity was previously reported, it seems that a reasonable strategy for revision would be to provide additional insight into the mechanism by which adiponectin is affecting adipocyte lipid uptake.

Specific suggestions: Cell biology: 1) Just showing the absence of vesicular structures and the loss of caveolin1 does not directly demonstrate that this is the mechanism of disrupted lipid uptake. So far this is correlative data. Do isolated adipocytes from the adiponectin KO/STZ mice or adiponectin KO/PANIC-ATTAC mice show poor uptake in the absence of insulin? Can adiponectin overcome such decreased activity? Are lipids actually in the vesicles that are defective in the experimental mice? Does the caveolin1 KO mouse show the same phenotype?

2) The studies showing increased ApoA-I staining in adipose tissue require clarification. The authors describe this as increased “lipoprotein” but they measure a single apolipoprotein and provide no information on lipids. There is no data presented showing that lipoproteins per se are increased in the vasculature. Do the authors see evidence of this by EM? ApoA-I is primarily carried on HDL, which does not have a major role in triglyceride delivery to adipose tissue. What about ApoE or ApoB? Is CD36 or SRB-I expression altered?

Signaling: 1) An important implication of the data is that insulin signaling is amplified by adiponectin under insulinopenic conditions. But there is no data to show p-AKT or other signaling nodes are indeed enhanced by insulin under these conditions. Is it this signaling pathway that is potentiated by adiponectin, or is it the lipolytic machinery that is more receptive to insulin suppression?

2) Through which receptors is adiponectin acting to mediate these effects? Are the effects of adiponectin on adipose responsiveness to insulin mediated through the classic adiponectin receptors through ceramide regulation?

---

## [Author Response]

*Cell biology: 1) Just showing the absence of vesicular structures and the loss of caveolin1 does not directly demonstrate that this is the mechanism of disrupted lipid uptake. So far this is correlative data. Do isolated adipocytes from the adiponectin KO/STZ mice or adiponectin KO/PANIC-ATTAC mice show poor uptake in the absence of insulin? Can adiponectin overcome such decreased activity? Are lipids actually in the vesicles that are defective in the experimental mice? Does the caveolin1 KO mouse show the same phenotype*?

To provide direct evidence as to how adiponectin modulates lipid uptake, we administrated BODIPY-labeled fatty acids to PANIC-ATTAC mice and imaged the BODIPY signal in subcutaneous adipose (new Figure 3 and Figure 3—figure supplement 2). Adiponectin KO exhibited significantly lower BODIPY accumulation in adipocytes compared to wildtype controls, while adiponectin transgenic overexpression dramatically increased the lipid uptake *in vivo*.

Although our EM images do not allow us to visualize lipids, caveolin-1 KO mice do indeed recapitulate the lipodystrophic phenotypes including lower body weight and fat mass, impaired triglyceride clearance, hypertriglyceridemia, adipocyte hypotrophy, and no change in lipoprotein lipase activity as demonstrated by our own, previously published papers (Razani et al., JBC, 2002; Wernstedt Asterholm et al., Cell Metabolism, 2012). These reports support the role of caveolin-1 in lipid endocytosis, and are discussed in the revised manuscript.

*2) The studies showing increased ApoA-I staining in adipose tissue require clarification. The authors describe this as increased “lipoprotein” but they measure a single apolipoprotein and provide no information on lipids. There is no data presented showing that lipoproteins per se are increased in the vasculature. Do the authors see evidence of this by EM? ApoA-I is primarily carried on HDL, which does not have a major role in triglyceride delivery to adipose tissue. What about ApoE or ApoB? Is CD36 or SRB-I expression altered*?

We performed immunofluorescence and quantitation of ApoB (new Figure 3 and Figure 3—figure supplement 4) and ApoE (new Figure 3 and Figure 3—figure supplement 5) to have an overview on the *in situ* distribution of apolipoproteins. In agreement with our observations on ApoA1, adiponectin KO exhibited stronger ApoB and ApoE signals associated with the vasculature marker endomucin compared to wildtype controls. We agree with the reviewer that apolipoproteins are only part of the lipoproteins, and our data on fractionation of circulating lipoproteins show increased triglyceride and cholesterol in VLDL in STZ-treated adiponectin KO relative to wildtype controls (Figure 2). The accumulation of apolipoproteins in local vasculature of adipose tissue is consistent with the lipid uptake deficiency in STZ-treated adiponectin KO, and may account for, at least in part, the elevation of VLDL lipid content observed.

Although we are not able to draw conclusions on vascular lipoproteins from the EM images, as suggested, we have examined the expression of CD36 (Figure 4) and fatty acid transport protein 1 (FATP1) (Figure 4) and observed no difference. Interestingly, expression of SR-B1 was increased in the subcutaneous WAT of STZ-treated adiponectin KO mice (new Figure 4), which could reflect a compensatory response for the accumulation of HDL component ApoA1 in circulation (Figure 3 and Figure 3—figure supplement 3).

*Signaling: 1) An important implication of the data is that insulin signaling is amplified by adiponectin under insulinopenic conditions. But there is no data to show p-AKT or other signaling nodes are indeed enhanced by insulin under these conditions. Is it this signaling pathway that is potentiated by adiponectin, or is it the lipolytic machinery that is more receptive to insulin suppression*?

We have analyzed the molecules downstream of insulin signaling in the subcutaneous WAT of STZ-treated mice (new Figure 4 and Figure 4—figure supplement 1). To mimic the insulinopenic conditions, mice were stimulated with a low dose of insulin (0.2 mU/g body weight). Akt phosphorylation was evident in wildtype mice, but completely abolished in adiponectin KO, where we also observed a trend towards reduced inhibition of HSL. Therefore, under insulinopenic conditions, adiponectin is required for insulin signaling, and supports the suppression of lipolysis.

*2) Through which receptors is adiponectin acting to mediate these effects? Are the effects of adiponectin on adipose responsiveness to insulin mediated through the classic adiponectin receptors through ceramide regulation*?

Investigation on the roles of adiponectin receptors awaits conditional knockout mouse models for both adiponectin receptors 1 and 2, to eliminate their compensatory effects on each other. This is discussed in the revised manuscript. The conventional knockout mouse models available in our laboratory result in non-viable double knockouts, with embryonic death between days 12 and 14 (our own unpublished observations).

Our laboratory has on-going efforts in comprehensive analyses of ceramides and other sphingolipids in multiple tissues under different pathophysiological conditions. We present a comprehensive survey of these lipid species in islets in our revised manuscript; however, we do not have adipose tissue available any more to determine adipocyte sphingolipid content.